# Food Security and Health Outcomes following Gray Divorce

**DOI:** 10.3390/nu16050633

**Published:** 2024-02-24

**Authors:** Hang Zhao, Tatiana Andreyeva, Xiaohan Sun

**Affiliations:** 1The Computer and Information Science Department, Allegheny College, 520 North Main Street, Meadville, PA 16335, USA; 2Department of Institutional Effectiveness, Allegheny College, 520 North Main Street, Meadville, PA 16335, USA; 3Department of Agricultural and Resource Economics, Rudd Center for Food Policy and Health, University of Connecticut, One Constitution Plaza, Hartford, CT 06103, USA; tatiana.andreyeva@uconn.edu; 4The Business and Economics Department, Allegheny College, 520 North Main Street, Meadville, PA 16335, USA; xsun@allegheny.edu

**Keywords:** gray divorce, food insecurity, depression, activities of daily living (ADLs)

## Abstract

The study evaluates the immediate and long-term consequences of gray divorce (i.e., marital dissolution after age 50) for the food security, depression, and disability of older Americans. Staggered Difference-in-Difference models were fitted to a nationally representative longitudinal sample of adults aged ≥ 50 years from the Health and Retirement Study, 1998–2018. Food insecurity and disability increase in the year of gray divorce and remain significantly elevated for up to six years or more following the event, consistent with the chronic strain model of gray divorce. Gray divorce has particularly adverse consequences for the food security of older women, while no gender differences were observed for disability. Increasing trends in gray divorce have important negative implications for food security and health of older Americans, particularly women, who appear to be less prepared to financially withstand a marital collapse in older age. Targeted policies to provide nutrition assistance and support in reemployment might be necessary to reduce the burden of food insecurity in the wake of gray divorce among women.

## 1. Introduction

As life expectancy improved dramatically in the 20th century, Western societies have gone gray. Today, one in six individuals in the United States is aged 65 or over, and this share is expected to grow [1]. Another phenomenon affecting many societies has been the growth in divorce rates, particularly in the US [2,3]. Gray divorce, a term referring to the dissolution of typically long-lasting marriages among people aged 50 years and above, has doubled in the US between 1990 and 2010 [4,5]. Although the increase in gray divorce has since slowed down among middle-aged adults (50–64 years), it has continued to climb among older adults (≥65 years of age) [5]. One of the most stressful life events, divorce, could have major negative implications for the well-being and health of an increasing number of older Americans. As such, the rising rates of gray divorce cannot be ignored, and policymakers need to understand the implications of gray divorce for diverse population groups to enable effective policy responses.

Prior research on divorce has differentiated its impact using a crisis model or a chronic strain model [6,7,8]. The chronic strain model suggests that divorce has lasting, perhaps permanent effects on individuals who endure chronic stresses following this major life crisis [9]. According to the classical sociological work of Durkheim [10] (1952), being married can ease economic burdens, support material well-being, and discourage unhealthy behaviors [11]. Thus, divorce results in a permanent negative effect by losing these economic and health-related advantages. In contrast, the crisis model emphasizes that the negative effects of divorce for most people are temporary, with full recovery to predivorce levels attained within a few years after the divorce. Compared with the benefits of being married and the chronic strains of being outside marriage, temporary stress caused by the transition in marital status could be the primary driver behind these outcome variations. Making the distinction between the short-term (crisis model) and the long-term (chronic strain model) effects of gray divorce is critical for providing evidence for relevant policy support. Studies of divorce have mixed results so far: some find evidence in support of the crisis model [12,13,14], while others believe the effects of divorce are better reflected by the chronic strain model [6,9,15,16].

Drawing from a nationally representative panel of older Americans with a range of rich variables, this study evaluates multiple outcomes of gray divorce from 1998 to 2018 and compares their fit to the crisis versus the chronic strain model. Whereas previous research focused on divorce in younger adulthood or measured single outcomes of marital dissolution, the current analysis offers a rigorous assessment of gray divorce in a nationally representative longitudinal study of older adults and includes multiple outcomes that are critical to the well-being of aging adults, such as food insecurity, depression, and disability.

Food insecurity, defined as limited access to adequate food due to lack of financial resources, disrupted the lives of 12.8 percent of U.S. households in 2022 [17]. Marital dissolution could have major negative implications for household food security as divorce often reduces household income and wealth due to the split of family resources, increased expenses of running a single-head household, and high legal costs of divorce. Economic implications of marital dissolution in older age could be even more severe than earlier in life, especially if remarriage or partnering does not occur and spouses have not accumulated substantial savings. Older Americans are experiencing decreased earnings. In 2013, the US annual median income was $24,644 for individuals aged between 65 and 74, compared to $38,643 for those aged 45 to 54 [18]. Baby boomers (born 1946–1964) and beyond generations have little retirement savings, and many of them rely on fixed incomes such as Social Security or Supplemental Security Income (SSI). These two programs account for at least half of the total income among adults aged over 65 and provide the only source of income for 24% of them [18]. Many baby boomers had to shift from traditional pension plans toward 401(k) plans and may have much lower savings due to a delayed start of 401(k) savings in life. According to a report from CNBC news in 2022, by age 75, individuals who had 401(k) plans had $86,000 less than those who had traditional pensions [19]. Older adults may be unable to rejoin the labor force or change career paths to sustain household income at predivorce levels, suggesting that divorce in older age may pose a much higher risk to economic security, including food security. A large share of the older population is at risk: over 15 million adults aged 65 and above had incomes below 200% of the federal poverty level [20], while 49% of adults aged 55–66 had no personal retirement savings at all in 2022 [21]. Any marital crisis for these people is likely to cause major negative effects, including poverty and food insecurity risk. Given the long-lasting economic impacts of divorce, particularly in the context of gray divorce, our hypothesis is that food insecurity resulting from gray divorce should be conceptualized within the framework of the chronic strain model. 

At the same time, depression has been identified as a significant problem for older adults, which can cause poorer performance in processing speed, verbal fluency, episodic memory, and other economic burdens on society [22,23,24]. Based on a systematic review and meta-analysis, 28.4% of older adults globally were found to have depression [25]. As gray divorce can be a traumatic event, post-divorce depression rates may increase, as shown in prior research reporting associations of worsening depressive symptoms with marital loss [26]. We therefore hypothesize that the occurrence of depression following gray divorce closely matches the expectations of the crisis model. The increase in depression symptoms right after a divorce is a direct result of the intense emotional pain from the marriage ending. However, the resilience and adaptive capabilities of older adults often may facilitate the recovery process, enabling them to regain their pre-divorce levels of psychological well-being within a few years [14]. 

Further, the prevalence of disability increases significantly with age [27], and the rates of new disability have climbed up in individuals aged 60 to 69 [28], potentially due to poor lifestyle choices including obesity, lack of physical activity, and increased stress [29]. As spouses often provide uncompensated care at home, marital collapse in older age could mean a loss of the primary caregiver for people with disabilities or at risk of developing disabilities. This vulnerable group of divorcees would have to navigate disability in old age on their own, facing the costly burden of hiring aid or receiving institutionalized care. Conversely, in the case of depression following gray divorce, which is hypothesized to align with the crisis model, the impact of gray divorce on Activities of Daily Living (ADLs)-based disability corresponds to the chronic strain model. We hypothesized that divorce leads to ongoing stress because one might lose a primary caregiver and then have to adjust to living alone or with additional (perhaps hired) support.

We use a difference-in-difference (DID) model to estimate average, immediate, and long-term changes in food security, depression, and ADL-based disability following gray divorce in a nationally representative panel of older Americans. The study evaluates whether implications of gray divorce for food security and health vary by gender and prior experience with divorce to provide evidence for targeted policy support. To our knowledge, this is the first assessment of how gray divorce is linked to food security among older Americans, which also tests the validity of the chronic strain and crisis models to describe gray divorce implications for health and well-being.

## 2. Materials and Methods

### 2.1. Data and Sample Selection 

The Health and Retirement Study (HRS) is an ongoing nationally representative longitudinal study of Americans ages 50 and above and their spouses, which has been tracking information about important components of older adults’ lives since 1992. Funded by the National Institute on Aging and with support from the Social Security Administration, the RAND HRS has collected rich longitudinal data to enable research in support of policies on retirement, healthcare insurance, savings, and economic well-being [30]. In the RAND HRS files, the observation unit is an individual respondent, and the financially knowledgeable household member also provides information about spouses [30]. Through biannual interviews and additional periodic supplements, the survey collects information about demographics, income and assets, physical and mental health, insurance, family transitions, healthcare utilization and costs, housing, labor force participation, and employment history. Additional details on the HRS are available elsewhere [30]. 

This study’s analytic sample is restricted to the HRS participants, with complete interviews between 1998 and 2018. Out of 42,234 individuals, our selection criteria targeted adults who were married upon their enrollment in the HRS panel and subsequently experienced divorce or separation from their spouse or partner between 1998–2018, while they were aged 50 or over (see Appendix A). Note that separation from living in a partnership is treated as divorce per standards in the literature [2,14]. Individuals who were not married/partnered at the beginning of the analytic sample were excluded. We further excluded observations with missing data, individuals who were never married, widowed, and remarried, leaving 736 divorced/separated individuals and 15,839 individuals who remained in marriages/partnerships throughout the analysis for the final analytical sample (101,486 individual-year observations). 

The exclusion of some groups from our analysis mitigates potential threats to the validity of our results, particularly concerning individuals who have experienced widowhood or remarriage. Considering widowhood as another significant life event that may have effects similar to those of gray divorce, its exclusion allows us to more precisely isolate the impacts attributable solely to gray divorce. This distinction is important, and widowhood’s effects are separately examined in the robustness tests section. Furthermore, the decision to exclude individuals who have remarried can avoid the complexities that come from the time between a first divorce and a subsequent remarriage. This period can confound the distinction between pre- and post-divorce effects. 

### 2.2. Measures

#### 2.2.1. Dependent Variables

Food insecurity was measured based on responses to two questions in the HRS questionnaire: (1) In the last two years, have you always had enough money to buy the food you need? and (2) At any time in the last two years, have you skipped meals or eaten less than you felt you should because there was not enough food in the house? Participants who answered affirmatively to at least one of the two questions were identified as food insecure. The severity of food insecurity was not assessed due to the limitations of the two available questions in the HRS. 

Disability assessment was based on a set of questions evaluating the functional limitations of activities of daily living (ADL) in six areas: bathing, dressing, eating, getting in/out of bed, walking across a room, and using the toilet [31]. Participants reporting some difficulty with performing at least one of these six daily tasks were identified as having functional limitations or experiencing ADL-based disability. 

The study relied on the Center for Epidemiologic Studies Depression (CES-D) Scale to assess mental health and screen for depression in the HRS participants [32]. Eight depression-related questions were asked, including six negative and two positive indicators. Negative indicators reflected the following symptoms: feeling depressed, feeling everything was an effort, sleep was restless, feeling alone, feeling sad, and could not get “going”. Positive indicators assessed if respondents felt happy and enjoyed life most of the time. If respondents reported four or more negative symptoms, they were classified as experiencing symptoms of depression [33]. Severity of depressive symptoms was not assessed. 

#### 2.2.2. Control Variables

A series of demographic and socioeconomic characteristics were added as covariates to improve the precision of the estimates, including time-variant variables such as age in years, age-squared, and household income. To account for the role of experiencing prior divorce in affecting the outcome of interest, we added an interaction term between the time since/to divorce and having had at least one divorce prior to entering our analysis. The variable was based on reports of the number of marriages/partnerships that a respondent has had in life. Individuals who experienced divorce/separation from their spouse or partner and had over one marriage/partner relationship before the analytic period of 1998–2018 were defined as having multiple experiences of divorce/separations. If individuals experienced at least one divorce, we defined a time-constant dichotomous variable distinguishing between individuals with the first divorce (coded as 0) and individuals who experienced another (perhaps second, third) divorce (coded as 1). In addition, we estimated all models by incorporating an interaction term between gender and the time since/to divorce, to test for possible heterogeneity in the outcomes of gray divorce for older men and women.

## 3. Identification Strategy

To estimate the effect of gray divorce on food insecurity, depression, and ADL-based disability, it is incorrect to simply compare these outcomes before and after gray divorce since they are not time-invariant variables and can change over time irrespective of any intervention, such as divorce. The difference-in-difference (DID) model can address this concern. Individuals in our analytic sample were all married/partnered at baseline in 1998 and experienced divorce/separation at some point during 2000–2018, introducing variation in the time of gray divorce, which allows us to use the DID model to identify causality. Under the assumption that gray divorce is random, the DID model can illustrate the causal effect of gray divorce on the outcomes of interest. To provide a comparison with individuals who remained in marriages/partnerships throughout the analysis, we included them in the control group of the DID model. 

The randomization of gray divorce is hard to identify. To address this, we exploited a propensity score to match individuals with gray divorce and individuals who remained in marriages throughout the analysis, based on gender, race, education, age, household income, and participation in the Supplemental Nutrition Assistance Program (SNAP) at baseline. Pierre-Carl and his colleagues developed this methodology in their study, estimating the effect of job loss on health [34]. We let $x_{k}$ be a vector of socioeconomic and demographic characteristics and $w_{k}$ be a variable of baseline SNAP participation for individual *k*, who experienced gray divorce. We let individual *j*, who remained in the marriage/partnership, be the closest neighbor of individual k. By defining the beginning wave (denoted by *s*), we can obtain a characteristic function for individual *k* and *j* at the beginning wave s: $q_{k,s}=(x_{ks},\;w_{ks})\;$ and $q_{j,s}=(x_{js},\;w_{js})$. We aimed to match individual *k* and *j* by exploiting the propensity score $ps\left(q_{k,s}\right)\;$ and $ps\left(q_{j,s}\right)$. Individual j is the nearest neighbor of individual *k* when ${min}_{j}|(ps(q_{k,s})-ps(q_{j,s})|\leq 0.001$. Figure 1 shows the overlap of propensity scores for stable partnerships and anticipated divorces at baseline. Eventually, we found 689 unique pairs in the matched groups.

Then, we applied the DID framework to estimate the average effects of gray divorce on the risk of food insecurity, depression, and ADL-based disability. The DID model can be specified as:(1)$$Y_{it}=\beta_{1}{Div}_{it}+\beta_{2}X_{it}+\alpha_{i}+\gamma_{t}+\varepsilon_{it}$$where $Y_{it\;}$is an outcome for respondent *i* in year *t*, ${Div}_{it}$ is a dichotomous measure of respondent *i* divorcing in year *t*, $X_{it}\;$ is a vector of explanatory variables such as age, quadratic age, and household income for respondent *i* in year *t*, $\alpha_{i}$ controls for individual fixed effects measuring time-invariant characteristics, $\gamma_{t}$ denotes wave fixed effects controlling for factors that change over time affecting all individuals, and $\varepsilon_{it}\;$ is the error term.

There would be a potential endogeneity problem if we simply compared individuals who remained in marriages/partnerships and individuals who experienced divorce/separation, since they are still quite different. For example, individuals who remained in marriages/partnerships might reside in more prosperous areas than individuals who experienced divorce/separation. Living environments and other unobservable factors could be associated with gray divorce, which also influences food insecurity and health outcomes. To overcome this concern, we limited the sample to individuals who were married at the time of joining the HRS panel but experienced divorce/separation from their spouse or partner at any point from 1998 to 2018. Individuals who remained in marriages/partnerships throughout the analysis were excluded.

The key premise of the DID model is the parallel trends assumption, which means that the trends of change in the control group are the same as the trends of change in the treatment group in the absence of the intervention. For example, if the increase in outcomes of the treatment group is faster than in the control group before the intervention, the DID model would overestimate the effect of gray divorce. To test the parallel trends assumption of the model and determine whether a break occurs after gray divorce, we exploited the time to and from divorce/separation to measure the experience of a marital collapse. Based on the reported marital status by each HRS respondent in every biannual interview (including married, married (spouse absent), partnered, separated, divorced, separated or divorced, widowed, and never married), we identified older adults who were divorced or separated in each two-year HRS period (i.e., wave) and tracked the exact wave of a reported divorce/separation based on the history of marital status self-reports. We then calculated the difference in the two-year intervals between each interview and the wave of divorce occurrence and created seven groups to describe the history of divorce: six or more years before divorce (lagged treatment indicator); four years before divorce (lagged treatment indicator); two years before divorce (reference group; lagged treatment indicator); the year/wave of divorce; two years after divorce; four years after; and six or more years after divorce. The statistical insignificance of these lagged treatment indicators before gray divorce indicates that the trends for the control and treatment groups are the same before the intervention. 

Furthermore, the event study specification can assess the prevalence of food insecurity, depression, and ADL-based disability before, during, and after a marital breakup. The model was specified as:(2)$$Y_{it}=\sum_{t\neq -2}\beta_{t}{SinceDiv}_{it}+{\delta X}_{it}+\alpha_{i}+\gamma_{t}+\varepsilon_{it}$$
where ${SinceDiv}_{it}\;$ is the categorical variable representing the number of years before or after divorce for individual *i* at year *t*, the coefficient of interest is $\beta_{t}$, which indicates the difference in the outcome between the reference group and other groups. We used two years before divorce as the reference group, so that all coefficients could be interpreted relative to the pre-divorce baseline. Using this period as the closest in time to the occurrence of divorce (i.e., intervention) can reduce concerns about any measurement errors that are farther apart between the reference group and the intervention.

To assess whether the outcomes of gray divorce vary by gender and experience of prior divorce, we added interaction terms between the time since/to divorce, gender, and prior divorce. The corresponding models are shown in Equations (3) and (4):(3)$$Y_{it}=\sum_{t\neq -2}\beta_{t}{SinceDiv}_{it}+{\delta_{1}X}_{it}+{Gender}_{i}\sum_{t\neq -2}\delta_{t}{SinceDiv}_{it}\;+\;\alpha_{i}+\gamma_{t}+\varepsilon_{it}$$
(4)$$Y_{it}=\sum_{t\neq -2}\beta_{t}{SinceDiv}_{it}+{\delta_{1}X}_{it}\;+priordivorce\sum_{t\neq -2}\delta_{t}{SinceDiv}_{it}\;+\;\alpha_{i}+\gamma_{t}+\varepsilon_{it}\;$$ In these models, gender and prior divorce were modeled as time-invariant characteristics. The coefficient of interest is $\delta_{t}\;$in Equations (3) and (4), which indicates the difference in estimates by gender and prior divorce status, respectively. All analyses were weighted to account for the HRS complex survey design to make estimates nationally representative of Americans aged 50 and above. All analyses were conducted in Stata 16 [35]. 

## 4. Results

Table 1 presents the baseline descriptive statistics for the 736 individuals who experienced gray divorce or separation within our sample. 

Table 2 provides a longitudinal view, tracing characteristics of the 736 individuals who experienced gray divorce over the 20-year study period. Importantly, while all 736 individuals underwent only one divorce during the analysis period, for 378 of these individuals, this event was not their first divorce.

Table 3 reports the predicted average effects of gray divorce on the probabilities of having food insecurity, depression, and ADL-based disability throughout the post-divorce period. For each period, the control group includes individuals who have not yet divorced (if they divorce in a future period) as well as those who remain married throughout the study period. These individuals serve as a comparison to those who experienced gray divorce in that same period. There is an average 4.4% (*p* < 0.01) increase in the probability of food insecurity after gray divorce, while depression risk increases on average by 3.5% (*p* < 0.10) and ADL-based disability by 5.7% (*p* < 0.01).

Table 4 provides estimates of the probability of food insecurity, depression, and ADL-based disability upon and after gray divorce. As compared to the baseline of two years prior to divorce, the risk of food insecurity among older adults increases by 7.3% in the year/wave of divorce (*p* < 0.01), by 9.5% in two years (*p* < 0.01), by 10.3% in four years (*p* < 0.01), and by 11.1% in six or more years, following the occurrence of divorce (*p* < 0.05). At the same time, the probability of ADL-based disability increases by 5.4% in the year of divorce (*p* < 0.05), by 7.9% (*p* < 0.05) in four years, and 8.7% (*p* < 0.1) in six or more years after gray divorce. No significant estimates were found on depression.

Figure 2 illustrates that, compared to two years before divorce, there are no significant differences in the time trend of the outcome measures before divorce, but significant increases were observed in the year of divorce for the probability of experiencing food insecurity and ADL-based disability. While no significant estimates were found for depression, older adults seemed to exhibit a trend towards returning to pre-divorce/separation levels of depression after their marital breakup, which aligns with predictions of the crisis model. However, increases in food insecurity and ADL-based disability after divorce persisted long-term, suggesting that the chronic strain model is more appropriate for describing these outcomes. 

Table 5 reports differences in the effects of gray divorce by gender, suggesting that women experience significantly more negative post-divorce changes in food security than men. For women, the probability of food insecurity increased rapidly around the year of gray divorce and remained at elevated levels in the post-divorce/separation period. Specifically, compared to men, the risk of food insecurity in women increased by 7.7% in the year of divorce (*p* < 0.1), 9.9% in the four years following divorce (*p* < 0.1), and 10.4% in six or more years since divorce (*p* < 0.1). At the same time, there were no significant gender differences in the risk of developing depression. There was a 7.5% higher probability of reporting ADL-based disability four years after divorce for older women compared to older men (*p* < 0.1). Appendix B presents the estimated changes in food insecurity separately by gender. 

To assess whether divorce impacts vary between individuals with more than one divorce and those experiencing their first divorce or separation, an interaction term was introduced. This term combined an indicator of more than one divorce in life with the trajectory of seven divorce categories (six or more years before divorce; four years before divorce; two years before divorce; the year/wave of divorce; two years after divorce; four years after divorce; and six or more years after divorce). Appendix B Table A1 shows results for food insecurity, depression, and ADL-based disability based on the occurrence of marriage. The interaction coefficients are not statistically significant for food insecurity, depression, and ADL-based disability, suggesting no differential effects for these outcomes for people experiencing their second, third, etc., divorce.

## 5. Robustness Tests

The study’s robustness check assesses how widowhood is linked to food insecurity and health and compares it with the effects of gray divorce. Widowhood is another important event in later life, often leading to poor health and potentially lower well-being. Anticipation and adaptation of bereavement can negatively affect widowers’ health [36]. However, compared with gray divorce, a widower could receive inheritance after the bereavement of the spouse. Thus, we hypothesize that widowhood does not affect food insecurity but has a negative effect on ADL-based disability. Table 6 shows estimation results, suggesting a small reduction in food insecurity at the year of the spouse’s death, which disappears overtime, and an increase in the risk of ADL-based disability, which remains over time.

## 6. Discussion

This study investigated the short- and longer-term impact of gray divorce on the onset of food insecurity, depression, and ADL-based disability overall, by gender and history of prior divorce. Utilizing a nationally representative panel of older Americans tracked over two decades, the study concluded that gray divorce leads to a sustained increase in the risk of food insecurity and ADL-based disability, aligning with the chronic strain model of divorce. While no significant effects were observed for depression post-divorce, an event study of depression identified a trend where depressive symptoms were alleviated within two years following marital dissolution.

The findings on food insecurity are in line with our hypothesis that divorce creates significant challenges to economic and food security due to reduced sources of income and the loss of savings of joint consumption, which are particularly relevant as people age out of the labor force and their health declines. Our findings are consistent with previous research indicating that being divorced increases the probability of food insecurity [37], but we quantify this risk for older adults specifically and establish the persistent, long-term nature of these negative shifts. The findings of this study, which point to the persistent nature of increased disability rates following gray divorce, align with previous research on divorce and health, particularly studies adopting a chronic strain model [38]. Similar to our findings from the event model, previous research indicated a full recovery from depression about a year following gray divorce [14]. 

We also find that gray divorce has particularly adverse consequences for food insecurity among older women, while men do not appear to be significantly affected. This is consistent with prior research suggesting that women face a higher risk of food insecurity relative to men [39]. This may be due to the traditional role of women in society, at least in the older generation, as homemakers and caregivers who are dependent financially on men. Divorce for older women could be a sudden event for which they are not mentally and financially prepared, creating major financial hardships. We did not observe any gender differences for depression and limitations in activities of daily living, which is consistent with prior findings [14]. There is also no evidence to suggest that the impact of gray divorce varies by the history of prior divorce. 

## 7. Strengths and Limitations

The study fills the gap in the literature on marital dissolution by focusing specifically on the older population and considering changes in multiple outcomes over a 20-year period to help estimate post-divorce changes in the wellbeing of older Americans. Previous studies on divorce often used cross-sectional samples or panel data for a short period of time [4,40,41]. Using a nationally representative longitudinal study that follows individuals before, during, and after gray divorce, we minimized potential measurement errors that are typically associated with cross-sectional studies. This enhanced the accuracy of our estimates and supported our capacity to distinguish the chronic strain model, especially evident in our findings related to food insecurity and ADLs. To our knowledge, this is also the first attempt to examine the chronic strain model for gray divorce and its implications for the food insecurity of older men and women. 

Along with its strengths, the study is subject to several limitations. First, the number of divorces in the subgroups is relatively low; thus, we only ran models on the overall sample of men and women when examining the role of prior divorce. It might consequently miss heterogenous effects of prior divorce on food insecurity and health outcomes for men and women. Further research with larger samples should examine this potential heterogeneity. Second, outcome measures were assessed based on the HRS questionnaires that rely on self-reports rather than objective measures such as doctors’ diagnoses of health conditions. Lack of data on the severity of food insecurity is another limitation in the assessment of gray divorce outcomes. Finally, despite the longitudinal design, the study cannot claim that the observed relationships are necessarily causal in nature because of the strong selection associated with divorce. At the same time, the strength of using 20-year longitudinal HRS data offsets a large proportion of the limitations and potential selection biases. Furthermore, using the DID framework leverages individual-level changes in gray divorce. Therefore, the study still provides a plausible causal assessment on the effects of gray divorce on food insecurity and health outcomes.

## 8. Conclusions

The study highlights a persistent increase in food insecurity among older adults—particularly women—following gray divorce, reflecting the chronic strain model of marital dissolution. We also discovered a significant and lasting negative impact on limitations of daily living activities from the year of gray divorce onwards, which further supports the chronic strain model over the crisis model.

These findings emphasize the necessity for targeted interventions. To alleviate food insecurity, strategies must be designed to the unique needs and vulnerabilities of older women undergoing divorce. They are particularly sensitive to such challenges due to potential health declines, limited employment opportunities, and constrained income. Further, policy frameworks and assistance programs must prioritize older adults, especially those under 65, as they often lack the additional support provided by federal programs available to those 65 and older, such as Medicare.

## Figures and Tables

**Figure 1 nutrients-16-00633-f001:**
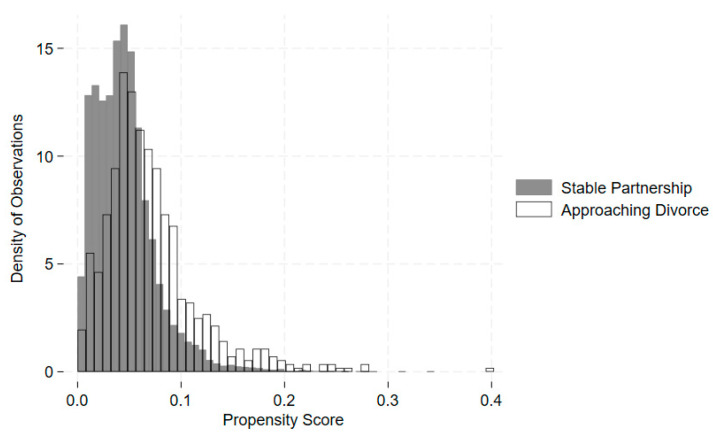
Comparison of propensity scores for stable partnerships and anticipated divorces at baseline. Note: The propensity scores displayed in the graph were computed using a logistic regression.

**Figure 2 nutrients-16-00633-f002:**
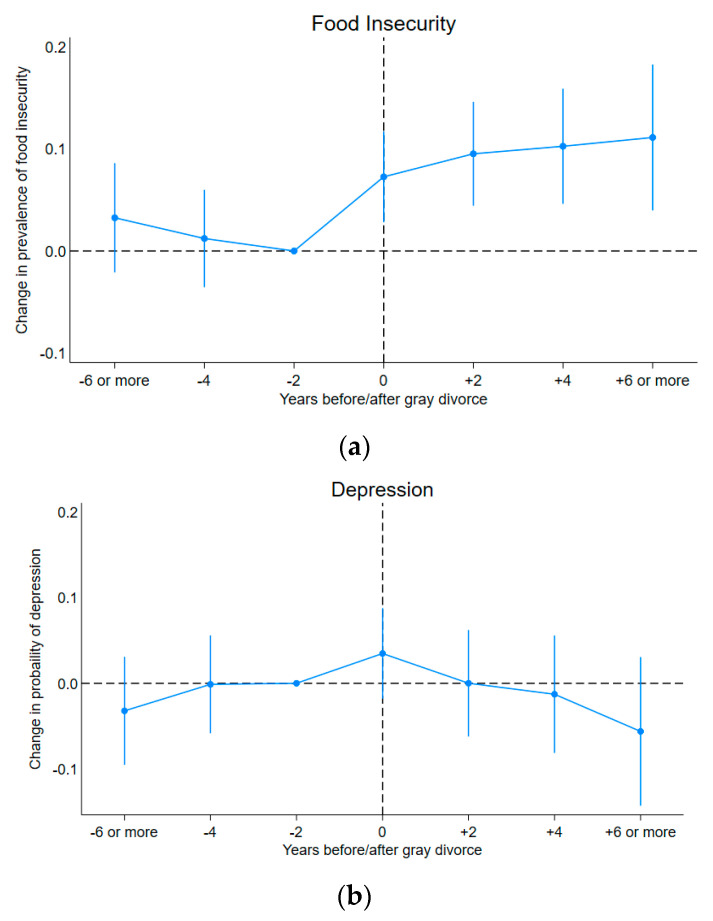
Predicted divorce-related changes in the prevalence of food insecurity, depression, and ADL ^1^-based disability relative to the reference category, based on Model 1 in Table 4. (**a**) Predicted changes in the prevalence of food insecurity before and after gray divorce. (**b**) Predicted changes in the prevalence of depression before and after gray divorce. (**c**) Predicted changes in the prevalence of ADL-based disability before and after gray divorce. Notes: Figures display point estimates and corresponding 95% confidence intervals. Two years before divorce (“−2”) is the reference category. ^1^ ADL: activities of daily living.

**Table 1 nutrients-16-00633-t001:** Descriptive statistics for baseline outcomes and key covariates, *n* = 736.

	% or Mean (SE)	Individual-Year Observations
Outcomes		
Food insecurity, %	16.2 (0.013)	
Depression, %	26.9 (0.016)	
ADL ^1^-based disability, %	18.1 (0.012)	
Demographic Characteristics		
Age	60.3 (0.325)	
Household income (log)	10.6 (0.046)	
Moderators (Time constant), %		
Women (vs. Men)	53.8	2429
Having more than one divorce in life (vs. first divorce)	52.2	2358
Number of divorcees		736

Notes: Analyses on outcomes and demographic characteristics were adjusted using the HRS survey weights. ^1^ ADL: activities of daily living.

**Table 2 nutrients-16-00633-t002:** Sample characteristics of participants in the HRS who got divorced/separated during 1998–2018.

	Divorced/Separated		Having more than 1 Divorce/Separation in Life		First Divorce/Separation		Women		Men	
	*n*	%	*n*	%	*n*	%	*n*	%	*n*	%
Years before/after gray divorce										
−6 or more	1062	23.5	557	23.6	505	23.4	561	23.1	501	24
−4	501	11.1	268	11.4	233	10.8	263	10.8	238	11.4
−2	665	14.7	346	14.7	319	14.8	343	14.1	322	15.4
0	736	16.3	378	16.0	358	16.6	383	15.8	353	16.9
2	486	10.8	268	11.4	218	10.1	264	10.9	222	10.6
4	374	8.3	198	8.4	176	8.1	207	8.5	167	8
+6 or more	695	15.4	343	14.5	352	16.3	408	16.8	287	13.7
Number of individuals	736		378		358		383		353	
Number of individual-year observations	4519	100	2358	100	2161	100	2429	100	2090	100

**Table 3 nutrients-16-00633-t003:** Average effects of divorce-related changes in the risk of food insecurity, depression, and ADL-based disability, *n* = 1378, consisting of individuals with a single neatest neighbor who remained married/partnered.

	Food Insecurity	Depression	ADL ^1^-Based Disability
	(1)	(2)	(3)
	Coef.	Coef.	Coef.
	(S.E.)	(S.E.)	(S.E.)
Divorce	0.044 ***	0.035 *	0.057 ***
	(0.015)	(0.019)	(0.016)
Age	−0.018	−0.006	−0.057 ***
	(0.011)	(0.016)	(0.014)
Age^2^	−0.000	0.0004 ***	0.0003 ***
	(0.000)	(0.0001)	(0.0001)
Household income (log)	−0.019	−0.008	−0.003
	(0.006)	(0.006)	(0.005)
R-squared	0.466	0.502	0.544
N. of individuals	1378	1378	1378

Notes: Robust standard errors. ^1^ ADL: activities of daily living; Data include individuals who remained in marriages/partnerships throughout the analysis. *** *p* < 0.01, * *p* < 0.1.

**Table 4 nutrients-16-00633-t004:** Staggered difference-in-difference model of divorce-related changes in the risk of food insecurity, depression, and ADL-based disability, *n* = 736.

	Food Insecurity	Depression	ADL ^1^-Based Disability
	(1)	(2)	(3)
	Coef. (S.E.)	Coef. (S.E.)	Coef. (S.E.)
Years before/after gray divorce (Ref. −2)			
−6 or more	0.033	−0.032	−0.031
	(0.032)	(0.037)	(0.031)
−4	0.012	−0.001	0.01
	(0.025)	(0.032)	(0.022)
0	0.073 ***	0.035	0.054 **
	(0.026)	(0.031)	(0.024)
2	0.095 ***	0.000	0.038
	(0.031)	(0.037)	(0.03)
4	0.103 ***	−0.013	0.079 **
	(0.037)	(0.042)	(0.036)
+6 or more	0.111 **	−0.056	0.087 *
	(0.046)	(0.055)	(0.049)
Age	−0.002	−0.028	−0.02
	(0.022)	(0.026)	(0.019)
Age^2^	0.000	0.000	0.000
	(0.000)	(0.000)	(0.000)
Household income (log)	−0.013	0.009	0.004
	(0.011)	(0.008)	(0.007)
R-squared	0.494	0.517	0.533
N. of individuals	736	736	736

Notes: Robust standard errors. All analyses were adjusted using the HRS survey weights. ^1^ ADL: activities of daily living. *** *p* < 0.01, ** *p* < 0.05, * *p* < 0.1.

**Table 5 nutrients-16-00633-t005:** Staggered difference-in-difference model of divorce-related changes in the risk of food insecurity, depression, and ADL-based disability by gender.

	Food Insecurity	Depression	ADL ^1^-Based Disability
	Coef. (S.E.)	Coef. (S.E.)	Coef. (S.E.)
Years before/after gray divorce (Ref. −2)			
−6 or more	0.024	−0.033	−0.024
	(0.044)	(0.049)	(0.039)
−4	0.025	−0.028	0.008
	(0.040)	(0.043)	(0.029)
0	0.035	0.002	0.025
	(0.030)	(0.043)	(0.033)
2	0.07 *	0.005	0.04
	(0.039)	(0.048)	(0.04)
4	0.053	−0.048	0.039
	(0.042)	(0.052)	(0.041)
+6 or more	0.054	−0.08	0.088
	(0.055)	(0.063)	(0.056)
Years before/after gray divorce (Ref. −2)			
* Women (vs. Men)			
−6 or more * Women	0.013	0.001	−0.013
	(0.051)	(0.063)	(0.045)
−4 * Women	−0.024	0.049	0.003
	(0.050)	(0.061)	(0.040)
0 * Women	0.077 *	0.064	0.056
	(0.046)	(0.056)	(0.043)
+2 * Women	0.054	−0.007	−0.003
	(0.055)	(0.061)	(0.047)
+4 * Women	0.099 *	0.068	0.075 *
	(0.053)	(0.059)	(0.046)
+6 or more * Women	0.104 *	0.045	0.003
	0.056)	(0.058)	(0.052)
Age	−0.007	−0.03 *	−0.021
	(0.022)	(0.026)	(0.019)
Age^2^	0.000	0.000	0.000 ***
	(0.000)	(0.000)	(0.000)
Household income (log)	−0.011	0.009	0.005
	(0.011)	(0.008)	(0.007)
R-squared	0.496	0.503	0.534
N. of individuals	736	736	736

Notes: Robust standard errors. All analyses were adjusted using the HRS survey weights. ^1^ ADL: activities of daily living. *** *p* < 0.01, * *p* < 0.1.

**Table 6 nutrients-16-00633-t006:** Falsification tests—staggered difference-in-difference model of widowhood in the risk of food insecurity and ADL-based disability.

	Food Insecurity	ADL ^1^-Based Disability
	(1)	(2)
	Coef. (S.E.)	Coef. (S.E.)
Years before/after gray divorce (Ref. −2)		
−6 or more	−0.001	0.006
	(0.007)	(0.01)
−4	0.009	−0.001
	(0.006)	(0.008)
0	−0.014 **	0.028 ***
	(0.006)	(0.008)
2	−0.006	0.018 *
	(0.008)	(0.01)
4	−0.003	0.027 **
	(0.008)	(0.013)
+6 or more	0	0.054 ***
	(0.009)	(0.015)
Age	−0.005	−0.048
	(0.005)	(0.008)
Age^2^	0	0
	(0)	(0)
Household income (log)	−0.011	−0.007
	(0.003)	(0.004)
R-squared	0.362	0.413
N. of individuals	3932	3933

Notes: Robust standard errors. All analyses were adjusted using the HRS survey weights. ^1^ ADL: activities of daily living. *** *p* < 0.01, ** *p* < 0.05, * *p* < 0.1.

## Data Availability

HRS data is accessible through the https://hrs.isr.umich.edu/data-products.

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
