# Peer review of "Food Security and Health Outcomes following Gray Divorce"

_nutrients, 2024, doi:10.3390/nu16050633_

Round 1
Reviewer 1 Report
Comments and Suggestions for Authors
The authors use a difference in difference design to estimate the impact of divorce in late life on multiple dimensions of wellbeing in late life using a large, nationally representative sample. The results are interesting and may have important implications. Nevertheless, there are several areas that could be improved.
First, I appreciate that multiple outcomes are considered (food insecurity, depression, ADL limitation). However, the authors stop short of developing the theoretical rationale to look at these. For instance, crisis models and chronic stress models are discussed, but it would be helpful to discuss how each model may fit each outcome.
Relatedly, it would be helpful to provide some context in the background section as to the relevant components of time for the outcomes (both levels of outcomes predivorce and potential lag in effects). For depression, it is likely that depressive symptoms may be elevated even years before couples ultimately divorce as they face marital conflicts that will be fatal to the marriage. However, this is not likely to be the case for food insecurity in which it is likely that very few partners would withhold food from a partner as they face marital conflicts. Regarding potential lags in effects, the theoretical background should discuss how long it may take for divorce to start impacting an outcome (food insecurity may be impacted almost immediately whereas effects on disability may take much longer).
Second, the authors make several exclusions to the sample and provide a detailed flow chart to show the sample selection. It would be helpful if the authors would take this a step further and discuss how exclusion of potentially relevant cases (e.g., exclusions due to missing data) may affect their results, or at a minimum, compare the characteristics of included versus excluded (eligible) respondents.
Similarly, I think that excluding those that remarry is a defensible decision, but effects of divorce during the years before remarriage could still be evaluated in those that remarry. This would help to approximate the overall effect of divorce in late-life rather than just the effects in which no remarriage is observed.
Third, regarding the definition of ADL limitations, the authors should discuss how those who respond "can't do" and "don't do" are treated in classifications.
Fourth, it would be helpful to provide a bit more context for Table 1 and Table 2, along with their differences. Is Table 1 (column labeled '% or mean") showing only characteristics at baseline? What is the sample size? Are only those who divorce being included? Similarly, in Table 2, should a column for persistently married respondents be shown? From my reading, the descriptive analyses never show the raw (unadjusted) comparisons of the divorced versus persistently married.
Fifth, the authors state on lines 234-235 that individual who remain married throughout the panel were excluded. This would mean that all individuals are divorced. From my understanding, DD estimators are based on differences in change in exposed versus unexposed persons. If all respondents are exposed to divorce, it may be unclear how this estimator is relevant.
Reviewer 2 Report
Comments and Suggestions for Authors
Thank you for the opportunity to read this paper on the immediate and long-term consequences of gray divorce (i.e., marital dissolution after age 50) for food security and health of older Americans. Food security in older adults is a neglected topic especially after gray divorce and yet of increasing importance and I acknowledge the authors for doing such studies. However, there are few minor issues that need to be addressed.
References have different formats. What is 28 on line 30.
Include the current national-level food insecurity percentage to facilitate group comparisons.
Line 101, in the introduction, briefly outline the DID model in two lines and/or provide a reference, as some readers may not be familiar with this statistical approach.
The Health and Retirement Study (HRS) is an ongoing nationally representative longitudinal study. However, the authors seem to have selected a distinct dataset for analysis, and the current study type is not specified. It may align with a retrospective cross-sectional study; if correct, please include.
I acknowledge that this involves secondary data; however, there should be a statement confirming consent from those permitting the use of their data in research. Unfortunately, the study lacks a consent or Research Ethics Board (REB) statement in the method section.
There is redundancy in the results presentation; the findings in Table 1 are reiterated in the first paragraph on page 5. If feasible, consider condensing the information.
On line 358, replacing "my findings" with "the study findings" would be more appropriate.
You've mentioned on page 3 the exclusion of specific groups like those never married, widowed, and remarried from the sample selection. However, the approach of testing robustness by comparing the effects of widowhood with gray divorce appears illogical for result or methodology validation. Essential details about this new subset group should be included, otherwise take this out. A more direct method is to compare both groups within your selected dataset or their subsets, offering clarity. Consider exploring alternative methods such as Subset Analysis, Sensitivity Analysis, and Cross-Validation for testing reliability and validity. Comparing your results with findings from other studies or external datasets focusing on similar research questions is a straightforward method for external validation, instilling additional confidence in result reliability.
There are few minor issues from flow to the structure of the sentences. There are some sentences missing articles and so on.
Overall, I think that is an important and interesting topic to explore policy changes affecting the food security of individuals impacted by gray divorce. The author/s has clearly devoted a great deal of time and effort to this study. I hope that these comments can help improve this study.
Comments on the Quality of English LanguageMinor issue. There are some sentences missing articles.
Round 2
Reviewer 1 Report
Comments and Suggestions for Authors
The authors have addressed my initial concerns and I have no further comments.